# Regulation of REM and Non-REM Sleep by Periaqueductal GABAergic Neurons

Franz Weber[1,3], Johnny Phong Hoang Do[1], Shinjae Chung[1,3], Kevin T. Beier[2], Mike Bikov[1], Mohammad Saffari Doost[1] & Yang Dan [1]

Mammalian sleep consists of distinct rapid eye movement (REM) and non-REM (NREM) states. The midbrain region ventrolateral periaqueductal gray (vlPAG) is known to be important for gating REM sleep, but the underlying neuronal mechanism is not well understood. Here, we show that activating vlPAG GABAergic neurons in mice suppresses the initiation and maintenance of REM sleep while consolidating NREM sleep, partly through their projection to the dorsolateral pons. Cell-type-specific recording and calcium imaging reveal that most vlPAG GABAergic neurons are strongly suppressed at REM sleep onset and activated at its termination. In addition to the rapid changes at brain state transitions, their activity decreases gradually between REM sleep and is reset by each REM episode in a duration-dependent manner, mirroring the accumulation and dissipation of REM sleep pressure. Thus, vlPAG GABAergic neurons powerfully gate REM sleep, and their firing rate modulation may contribute to the ultradian rhythm of REM/NREM alternation.

[1] Division of Neurobiology, Department of Molecular and Cell Biology, Helen Wills Neuroscience Institute, Howard Hughes Medical Institute, University of California, Berkeley, CA 94720, USA. [2] Department of Biology, Howard Hughes Medical Institute, Stanford University, Stanford, CA 94305, USA. [3] Present address: Department of Neuroscience, Perelman School of Medicine, University of Pennsylvania, Philadelphia, PA 19104, USA. Franz Weber and Johnny Phong Hoang Do contributed equally to this work. Correspondence and requests for materials should be addressed to Y.D. (email: ydan@berkeley.edu)

During behavioral sleep, the mammalian brain alternates between two distinct states—rapid eye movement (REM) sleep with desynchronized electroencephalogram (EEG) and non-REM (NREM) sleep characterized by large-amplitude slow-wave activity[1, 2]. While the switch between these these brain states occurs rapidly within seconds, the recurrence of the NREM/REM cycle follows a much slower ultradian rhythm, on a timescale of minutes in rodents to hours in humans[3]. Understanding the mechanisms controlling sleep requires characterization of the neuronal processes operating on both timescales.

The ventrolateral periaqueductal gray (vlPAG) in the midbrain is known to play an important role in gating REM sleep. Inhibition of the vlPAG and the nearby deep mesencephalic reticular nucleus through muscimol injection causes a marked increase in REM sleep[4–7], whereas disinhibition of this area with bicuculline decreases REM sleep. Lesion of the vlPAG neurons also increases REM sleep in cats[8], rats[9], and mice[10]. c-Fos immunohistochemistry following REM sleep deprivation identified a large number of REM-off neurons that are GABAergic[7], and recent studies based on optogenetic[11] and chemogenetic[12] manipulations demonstrated that activation/inhibition of the vlPAG GABAergic neurons decreases/increases REM sleep.

Although these studies indicate a crucial role of the GABAergic neurons in gating REM sleep, the underlying circuit mechanism is not well understood. Anatomical tracing revealed a strong vlPAG projection to the dorsolateral pons[13], an important REM-promoting area[9, 14–17], but the functional contribution of this projection in gating REM sleep remains to be assessed. The onset and termination of each REM sleep episode occur rapidly on a timescale of seconds. Although vlPAG GABAergic neurons have been shown to express c-fos after REM sleep deprivation[7], the slow time course of c-fos expression obscures their firing rate changes associated with brain state transitions. Furthermore, the REM/NREM alternation is known to follow an ultradian rhythm on a timescale of minutes, but the mechanism controlling the ultradian timing is poorly understood. Given the crucial role of vlPAG GABAergic neurons in gating REM sleep, an interesting question is whether their activity is modulated on a timescale of minutes and whether such slow modulation could influence the ultradian timing of REM sleep.

In this study, we characterize the activity of vlPAG GABAergic neurons on both fast and slow timescales and assessed their roles in regulating REM and NREM sleep. We first measure the effects of activating these neurons on brain states. Rabies virus (RV)-mediated trans-synaptic retrograde tracing is used to establish the postsynaptic targets of these neurons in the dorsolateral pons, and the functional contribution of this projection in gating REM sleep is tested optogenetically. We then perform cell-type-specific recording and calcium imaging to characterize the spiking activity of these GABAergic neurons across sleep–wake cycles. Analysis of their firing rate modulation show that, while the rapid activity changes over seconds are directly associated with REM sleep onset and termination, their slow modulations over a time course of minutes closely mirror the accumulation and dissipation of REM sleep pressure, well suited for the ultradian regulation of the REM/NREM alternation. These results clarify the roles of vlPAG GABAergic neurons in both the direct gating and ultradian regulation of REM sleep.

## Results

**Optogenetic activation of vlPAG GABAergic neurons**. Our previous study showed that channelrhodopsin 2 (ChR2)-mediated optogenetic activation of vlPAG GABAergic neurons (20 Hz laser stimulation, 5 min per trial, applied every 15–25 min) causes a near-complete suppression of REM sleep, a strong reduction of wakefulness, and a marked increase in NREM sleep[11]. We have now confirmed these effects in additional animals (Fig. 1a–c, Supplementary Fig. 1a, $P < 0.0001$ for all three brain states; bootstrap, $n = 12$ mice). The EEG power spectrum during spontaneous NREM sleep was indistinguishable from that during the NREM sleep overlapping with laser activation of ChR2 (Supplementary Fig. 1b), and in control mice expressing enhanced yellow fluorescent protein (eYFP) without ChR2 laser stimulation had no effect on brain states (Supplementary Fig. 1c). Histological examination confirmed ChR2-eYFP expression and optic fiber placement in the vlPAG (Supplementary Fig. 1d, e), and across mice the effect of optogenetic stimulation was highly consistent (Supplementary Fig. 1f, g). Furthermore, ChR2-medicated activation of glutamatergic neurons in the vlPAG strongly promoted wakefulness (Supplementary Fig. 2), distinct from the effect of activating GABAergic neurons.

To distinguish whether the laser-induced change in the percentage of time in each brain state is due to a change in the initiation or maintenance of that state, we analyzed the transition probability between each pair of brain states. Laser activation of the vlPAG GABAergic neurons caused marked decreases in NREM→REM and REM→REM transitions (Fig. 1d–f; $P < 0.001$, bootstrap), indicating a suppression of both the initiation and maintenance of REM sleep. Laser stimulation also caused significant increases in NREM→NREM and wake→NREM transitions ($P < 0.001$). However, as shown in the cumulative transition probability analysis, the increase in the wake→NREM transition was much weaker compared to the decreases in NREM→wake and NREM→REM transitions (Fig. 1f), indicating that the increased NREM sleep was primarily due to enhanced maintenance rather than initiation. Consistent with these changes in transition probabilities, the mean duration of REM episodes was shortened ($P = 0.04$, $z = 7.0$, Wilcoxon sign-rank test), and the NREM episodes were prolonged (Fig. 1g; $P = 0.004$, $z = 2.0$). During the baseline period without laser, 22.2% of the NREM episodes transitioned into REM sleep and 77.8% to wakefulness; laser stimulation significantly reduced the proportion of transitions into REM sleep to 7.6% (Fig. 1h; $P = 0.003$, $z = 1.0$, Wilcoxon signed-rank test). Interestingly, the effect of laser stimulation on the initiation of wakefulness depended on the preceding state: while the NREM→wake transition was reduced ($P < 0.001$), the REM→wake transition was enhanced (Fig. 1d, e; $P = 0.049$). Thus, unlike the unambiguous suppression of REM sleep by vlPAG GABAergic neurons, their effect on wakefulness is more complex: wakefulness is enhanced relative to REM sleep but suppressed relative to NREM sleep, with an overall reduction in the amount of wakefulness.

Previous studies have shown that lesion of the vlPAG increases REM sleep[8–10]. To test the effect of cell-type-specific ablation of GABAergic neurons, we injected into the vlPAG of GAD2-Cre mice Cre-inducible adeno-associated viruses (AAV) expressing pro-Caspase 3 and an enzyme that cleaves pro-Caspase 3 into its active, pro-apoptotic form Caspase 3[18]. Compared to the control GAD2-Cre mice expressing eYFP, the number of GAD2-expressing neurons in the vlPAG was greatly reduced (Supplementary Fig. 3a, b). The amount of REM sleep increased significantly during both the light ($P = 0.009$, $z = 50.0$, Wilcoxon rank-sum test) and dark phases ($P = 0.03$, $z = 0.26$; Supplementary Fig. 3c–i), which was due primarily to an increased frequency of REM sleep episodes (light phase, $P = 0.02$, $z = 0.49$; dark phase, $P = 0.03$, $z = 0.26$; Supplementary Fig. 3f, i). This is consistent with previous lesion studies[8–10], and it further supports the role of vlPAG GABAergic neurons in gating REM sleep.

**Function of vlPAG projection to dorsolateral pons**. Previous studies suggested that the vlPAG inhibits the dorsolateral pons,

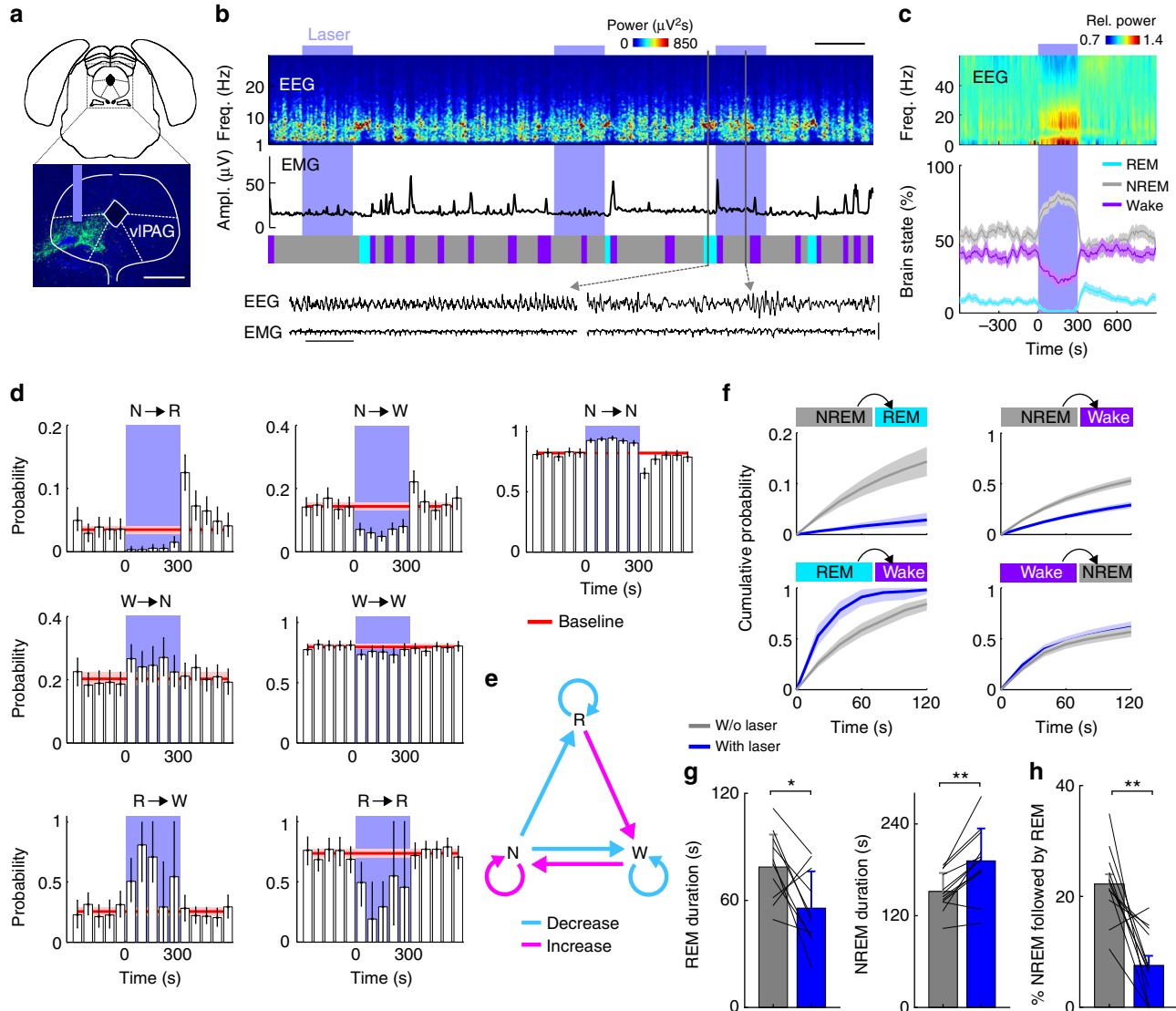

**Fig. 1** Optogenetic activation of vlPAG GABAergic neurons suppresses REM sleep and wakefulness while enhancing NREM sleep. **a** Schematic of optogenetic experiment. Top, coronal diagram of mouse brain; bottom, fluorescence image of PAG in a *GAD2-Cre* mouse injected with AAV expressing ChR2–eYFP (green). Blue, 4′,6-diamidino-2-phenylindole (DAPI). Scale bar, 500 μm. Brain figure adapted from Allen Mouse Brain Atlas (© 2015 Allen Institute for Brain Science. Allen Brain Atlas API. Available from: http://brain-map.org/api/index.html). **b** Example experiment. Shown are EEG power spectrogram (scale bar, 300 s), electromyogram (EMG) amplitude, brain states, and EEG, EMG raw traces on an expanded timescale during the selected time periods (black boxes; scale bars, 1 s and 0.5 mV). Blue shading, laser stimulation period (20 Hz, 300 s). **c** Average EEG spectrogram (top, normalized by the mean power in each frequency band) and the percentage of wake, NREM, or REM states (bottom) before, during, and after laser stimulation ($n = 12$ mice). Laser stimulation increased NREM sleep ($P < 0.0001$, bootstrap) and decreased wakefulness ($P < 0.0001$) and REM sleep ($P < 0.0001$). Shading, 95% confidence intervals (CI). Blue stripe, laser stimulation period (20 Hz, 300 s). **d** Effect of laser stimulation on transition probability between each pair of brain states. Bars, transition probabilities within each 20 s period. Error bar, 95% CI (bootstrap). Red line and shading, baseline and 95% CI. N NREM, R REM, W wake. **e** Diagram summarizing transition probabilities that are significantly increased (magenta) or decreased (cyan) by laser stimulation. **f** Cumulative probability of transition between each pair of brain states within 120 s after the initiation of each brain state with or without laser stimulation. Laser stimulation caused a decrease in NREM→REM ($P < 0.001$, bootstrap) and NREM→wake ($P < 0.001$) transitions, increase in REM→wake transition ($P < 0.001$), and only minor increase in wake→NREM transition ($P = 0.028$). Shading, 95% CI. **g** Mean REM and NREM episode duration for episodes overlapping or non-overlapping with laser ($n = 12$ mice). Lines, single mice. *$P < 0.05$, Wilcoxon signed-rank test. Error bar, ±s.d. (**h**) Percentage of NREM episodes followed by REM, in episodes overlapping or not overlapping with laser. *$P < 0.05$, **$P < 0.01$, Wilcoxon singed-rank test. Error bar ±s.d.

including the sublaterodorsal nucleus (SLD)[9, 13], known to be crucial for the generation of REM sleep. However, the functional contribution of this inhibitory projection has not been tested directly. We injected Cre-inducible AAV expressing ChR2–eYFP into the vlPAG of *GAD2-Cre* mice and implanted an optic fiber near the ventrolateral boundary of the pontine central gray to stimulate axon fibers projecting to this region of the dorsolateral pons (Fig. 2a, Supplementary Fig. 4a–c). Activation of the vlPAG

axons in this region caused a near-complete suppression of REM sleep and strong reduction of wakefulness (Fig. 2b; $P < 0.0001$, bootstrap). The magnitudes of the effects were comparable to those caused by stimulating the vlPAG cell bodies (Fig. 1c), suggesting a strong functional contribution of this projection.

Glutamatergic neurons in the SLD are known to be important for REM sleep generation[9, 14–17]. To test whether these neurons are directly innervated by the vlPAG GABAergic projection, we

used RV-mediated trans-synaptic retrograde tracing[19–23]. A mutant version of the avian-specific retroviral TVA receptor fused with mCherry (TC[66T]) and rabies glycoprotein (RG) were expressed by injecting two Cre-inducible AAV vectors (AAV2-*CAG-FLEx-TC[66T]* and AAV8-*CAG-FLEx-RG*) into the SLD

region of *VGLUT2-Cre* mice. A modified RV expressing enhanced green fluorescent protein (eGFP) (RV*dG*-eGFP+EnvA) was injected 2 weeks later at the same site to infect the cells expressing TC[66T] and label their presynaptic inputs (Fig. 2c, d). To test whether presynaptic, eGFP-labeled neurons in the vlPAG

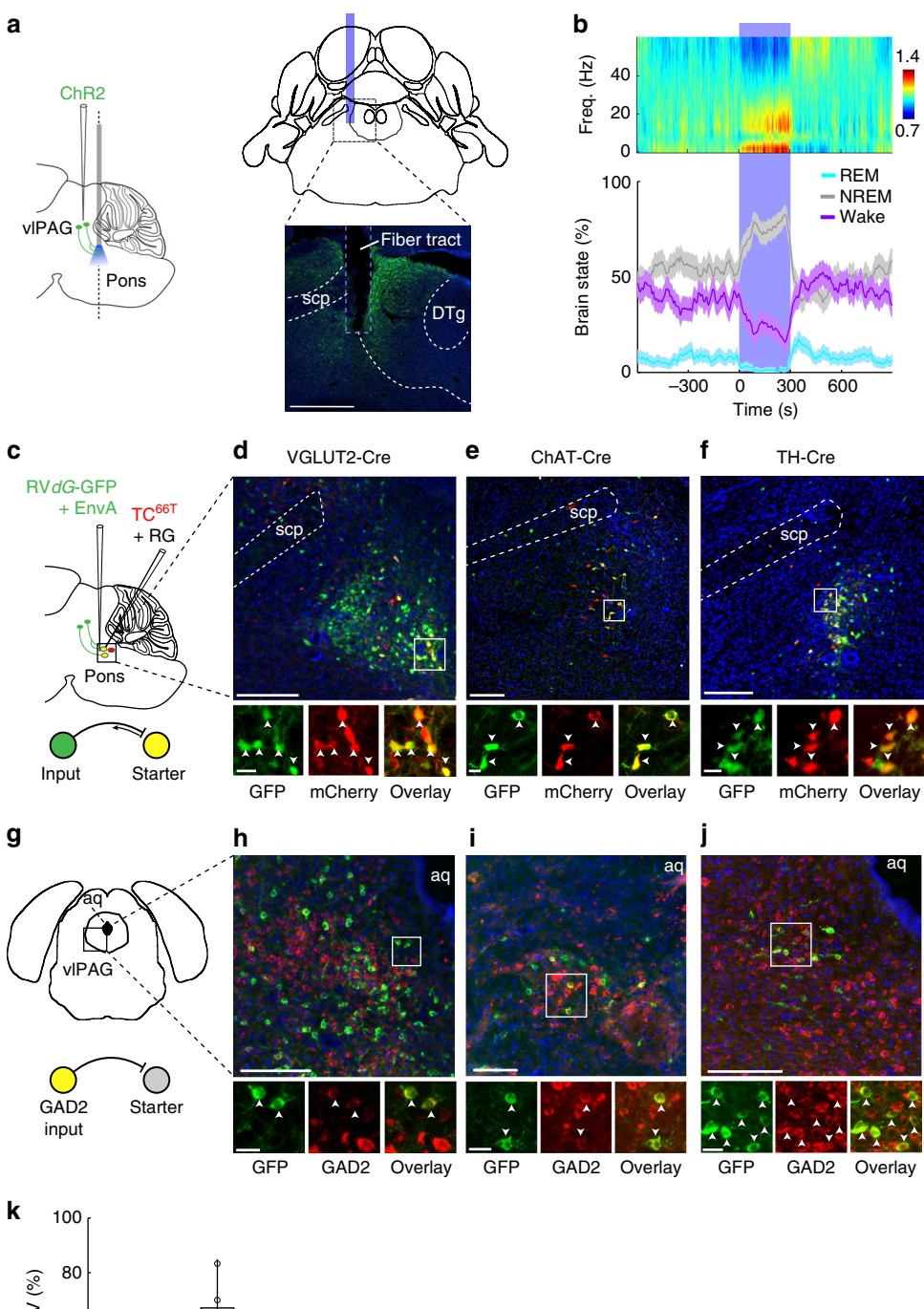

are GABAergic, we performed in situ fluorescence hybridization (FISH) for *GAD2* mRNA (see Methods section). We found that 41% (317/782) of eGFP-labeled vlPAG neurons were GABAergic (Fig. 2g, h, k; *n* = 3 mice), suggesting that the suppression of REM sleep by vlPAG GABAergic neurons could be partly mediated by their direct inhibition of SLD glutamatergic neurons.

Since the cholinergic laterodorsal tegmental nucleus (LDT) and noradrenergic locus coeruleus (LC) are also within the projection field of the vlPAG GABAergic neurons, we tested their monosynaptic innervation using RV-mediated retrograde tracing. We found that 34% (33/96) of eGFP-labeled vlPAG neurons innervating cholinergic LDT neurons were GABAergic (Fig. 2e, i, k; *n* = 3 mice), and 56% (83/148) of the neurons innervating noradrenergic LC neurons were GABAergic (Fig. 2f, j, k; *n* = 3 mice). Stimulating the LC noradrenergic neurons increases wakefulness[24]. Optogenetic activation of cholinergic neurons in the LDT and the nearby pedunculopontine tegmentum increased NREM to REM transitions[25], although chemogenetic activation of these neurons was found to promote light NREM sleep instead[26]. Our result suggests that the suppression of REM sleep/wakefulness and consolidation of NREM sleep by vlPAG GABAergic neurons could also be mediated in part by inhibition of the cholinergic and noradrenergic neurons in dorsolateral pons.

**Optrode recording from vlPAG GABAergic neurons.** To understand how vlPAG GABAergic neurons control the initiation and termination of REM sleep, it is important to characterize their firing rate changes at brain state transitions. Although a powerful method for identifying REM-on and REM-off neurons[7,9,15], c-Fos immunohistochemistry cannot detect activity changes on the timescale of seconds, given the slow time course of c-fos expression. While electrophysiological recordings have been made in the vlPAG previously[6,27,28], it is unclear whether the recorded neurons were GABAergic.

To identify vlPAG GABAergic neurons, we used optogenetic tagging and optrode recording[29]. High-frequency laser pulse trains (15 or 30 Hz, 5–10 ms/pulse, 1 or 0.5 s/train) were applied in the vlPAG of *GAD2-Cre* mice injected with Cre-inducible AAV-expressing ChR2-eYFP. Single units exhibiting reliable laser-evoked spiking with short latencies and high reliability were identified as GABAergic neurons (Fig. 3a and Supplementary Fig. 5a, b; see Methods section), which typically had biphasic spike waveforms (Supplementary Fig. 5c). Laser stimulation increased the firing rates of all identified units, irrespective of their baseline activity (Supplementary Fig. 5d, e), and subsequent

histological examination confirmed the locations of the identified units (see Methods section, Supplementary Fig. 5f). The majority of these neurons (11/19, 58%) were significantly less active during REM sleep than during both NREM sleep and wakefulness (P < 0.05, Wilcoxon rank-sum test, post-hoc Bonferroni correction, Fig. 3b–d, Supplementary Movie 1), consistent with the large number of REM-off GABAergic neurons detected by c-Fos immunohistochemistry[7]. A small number of identified neurons (4/19, 21%) were most active during REM sleep, consistent with the existence of REM-on neurons in this region[6,7,10,28]. During wakefulness, vlPAG GABAergic neurons were preferentially active during locomotor behaviors (Supplementary Fig. 5g). Compared to the identified GABAergic neurons, the unidentified cell population contained a lower percentage of REM-off neurons (Supplementary Fig. 6a–d; 4/22, 18.2%). Interestingly, some neurons in the vlPAG were inhibited by the laser pulses, suggesting that they receive inhibitory inputs from vlPAG GABAergic neurons. This neuronal population contained the highest percentage of REM-on neurons (6/10, 60%; Supplementary Fig. 6c–f), suggesting that local inhibition of REM-on neurons within the vlPAG also contributes to REM sleep suppression by the GABAergic neurons.

We then analyzed the firing rate changes of these neurons at brain state transitions. The GABAergic REM-off neurons showed a strong firing rate reduction seconds before the onset of REM sleep, which persisted through most of the REM sleep episode (Fig. 3e, Supplementary Fig. 7a), suggesting that the silencing of these neurons contributes to natural REM sleep onset and maintenance. In contrast, the REM→wake transition was associated with an abrupt increase in firing rate, consistent with the finding that optogenetic activation of these neurons shortened REM sleep by increasing the transition into wakefulness (Fig. 1d–f). Surprisingly, the NREM→wake transition was also accompanied by a firing rate increase and the wake→NREM transition by a decrease (Fig. 3e, see Discussion section). Interestingly, while their firing rates decreased gradually over the course of each NREM sleep episode before the transition to either REM sleep or wakefulness, the overall firing rate was significantly lower during the episodes leading to REM sleep (Fig. 3f; $P = 0.0001$, $T(8) = -7.03$, paired *t*-test; Supplementary Fig. 7b). This is consistent with the finding that NREM episodes with optogenetic activation of the GABAergic neurons were less likely to transition into REM sleep (Fig. 1h); it also suggests that the natural firing rates of vlPAG GABAergic neurons during each NREM episode is predictive of whether the animal will transition into REM sleep or wakefulness. To test this possibility directly, we predicted the transition to either REM sleep or wakefulness based

**Fig. 2** Inhibition of dorsolateral pons by vlPAG GABAergic projection suppresses REM sleep and wakefulness while promoting NREM sleep. **a** Left, schematic depicting ChR2-mediated activation of GABAergic axons projecting from vlPAG to the dorsolateral pons. Right, coronal diagram of mouse brain (top) and fluorescence image (bottom) of dorsolateral pons (black box in diagram) in a *GAD2-Cre* mouse injected with AAV expressing ChR2–eYFP (green) into the vlPAG. Blue bar, optic fiber. Blue, DAPI. Scale bar, 500 μm. DTg dorsal tegmental nucleus, scp superior cerebellar peduncle. **b** Top, normalized EEG spectrogram. Bottom, percentage of REM, NREM, or wake states before, during, and after laser stimulation of GABAergic axons (*n* = 5). Shading, 95% CI. **c** Schematic showing rabies-mediated trans-synaptic tracing. TC[66T] mutant EnvA receptor fused with mCherry, RG rabies glycoprotein, RV*dG* RG-deleted rabies virus. **d** Top, fluorescence image of SLD in *VGLUT2-Cre* mouse. Scale bar, 200 μm. Bottom, enlarged view of white box showing starter cells (yellow, arrowheads), expressing both eGFP and mCherry. Scale bar, 20 μm. **e** Top, fluorescence image of LDT in *ChAT-Cre* mouse. Scale bar, 200 μm. Bottom, enlarged view of white box showing starter cells (yellow, arrowheads). Scale bar, 20 μm. **f** Top, fluorescence image of LC in *TH-Cre* mouse. Scale bar, 200 μm. Bottom, enlarged view of white box showing starter cells (yellow, arrowheads). Scale bar, 20 μm. **g** Coronal diagram of mouse brain. Aq aqueduct. **h** vlPAG GABAergic (*GAD2*) neurons innervating glutamatergic (*VGLUT2*) SLD neurons. Top, fluorescence image showing rabies-eGFP-labeled (green) and *GAD2*-positive neurons in vlPAG (*n* = 3 mice). Red, *GAD2*. Blue, DAPI. Scale bar, 200 μm. Bottom, enlarged view of white box showing eGFP-labeled neurons expressing *GAD2* (arrowheads). Scale bar, 30 μm. **i** Similar to **h**, for vlPAG GABAergic neurons innervating cholinergic (*ChAT*) LDT neurons (*n* = 3). **j** Similar to **h**, for vlPAG GABAergic neurons innervating noradrenergic (*TH*) LC neurons (*n* = 3). **k** Percentages of rabies-eGFP-labeled neurons in vlPAG innervating glutamatergic SLD neurons, cholinergic LDT neurons, and noradrenergic LC neurons (*n* = 3). Circles, single mice; error bars, ±s.d. Brain figures in **a**, **c**, **g** were adapted from Allen Mouse Brain Atlas (© 2015 Allen Institute for Brain Science. Allen Brain Atlas API. Available from: http://brain-map.org/api/index.html)

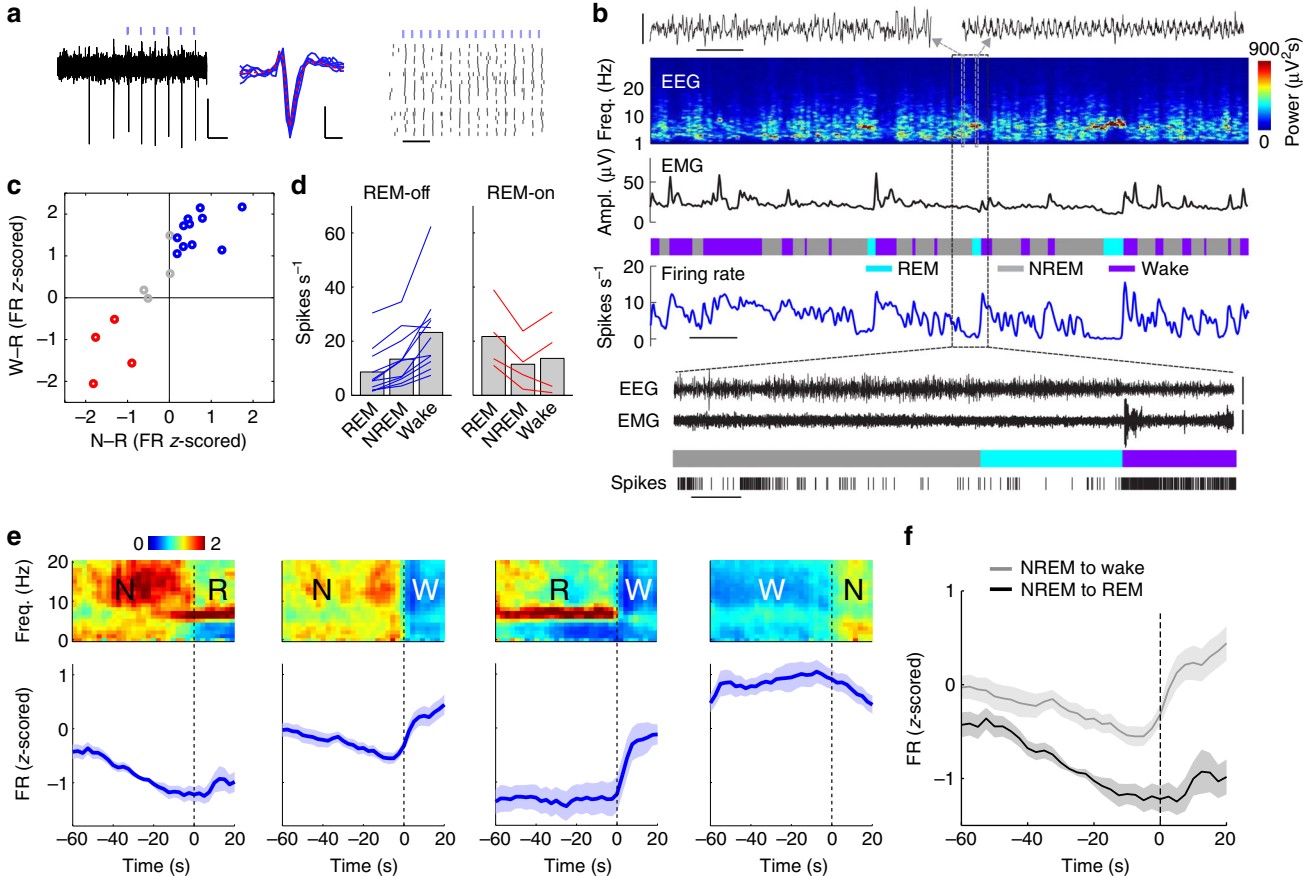

**Fig. 3** Firing rates of identified vlPAG GABAergic neurons across brain states. **a** Example unit. Left, raw trace showing spontaneous and laser-evoked spikes. Blue ticks, laser pulses (15 Hz). Scale bars, 100 ms, 0.5 mV. Middle, comparison between laser-evoked (blue) and averaged spontaneous (red) spike waveforms from this unit. Scale bars, 0.2 ms, 0.5 mV. Right, Spike raster showing multiple trials of laser stimulation at 30 Hz. Scale bar, 100 ms. **b** Firing rates of an example vlPAG GABAergic neuron (blue) along with EEG spectrogram, EMG amplitude, and color-coded brain state (scale bar, 120 s). Two example EEG raw traces (indicated by gray boxes) are shown on top of the EEG spectrogram (scale bars, 1 s, 0.5 mV). The timing of single spikes (vertical ticks) is depicted on an expanded timescale (indicated by black box) along with EEG, EMG raw traces (scale bars, 10 s, 0.5 mV). **c** Firing rate modulation of 19 identified units from 6 mice. W wake, R REM, N NREM. Blue, significant REM-off neurons (P < 0.05, Wilcoxon rank-sum test, post-hoc Bonferroni correction); red, significant REM-on neurons; gray, other neurons. **d** Firing rates of significant REM-off (left) and REM-on (red) neurons during different brain states. Each line shows firing rates of one unit; gray bar, average across units. **e** Average EEG spectrogram (upper, normalized by the mean power in each frequency band) and mean firing rate (z-scored) of significant REM-off neurons (lower) at brain state transitions. Shading, ±s.e.m. **f** Firing rates during NREM episodes preceding wake were significantly higher than those preceding REM episodes (P = 0.0001, T(8) = −7.03, paired t-test)

on the preceding NREM-period activity of each REM-off neuron using a linear classification method (Fisher's linear discriminant; see Methods section). We found that even 60 s before the transition the prediction accuracy was significantly above the chance level (P = 0.04, bootstrap), and it reached 68.6% 30 s before the transition (Supplementary Fig. 8a). Thus the NREM activity of individual vlPAG REM-off GABAergic neurons is informative about future brain state changes. In contrast, the REM-on neurons in the unidentified population showed no firing rate difference between the NREM episodes leading to REM or wakefulness (Supplementary Fig. 6g, h; P = 0.31, T(10) = 1.06, paired t-test; Supplementary Fig. 8b).

**Calcium imaging of vlPAG GABAergic neurons.** In addition to optrode recording, we also performed calcium imaging, which allows observation of multiple vlPAG GABAergic neurons simultaneously. AAV was injected into the vlPAG of *GAD2-Cre* mice for Cre-inducible expression of the calcium indicator GCaMP6f[30]. Imaging was performed in freely moving mice through a gradient refractive index (GRIN) lens coupled to a

miniaturized integrated fluorescence microscope[31] (Fig. 4a, Supplementary Fig. 9).

We observed strong variation of the calcium activity across brain states (Fig. 4b, Supplementary Movie 2). The great majority of imaged neurons exhibited lower activity during REM sleep than NREM sleep or wakefulness (Fig. 4c, d). Analysis at brain state transitions showed a strong activity suppression before the NREM→REM transition and throughout the REM sleep episode but a rapid increase in activity at wake onset (Fig. 4e). Furthermore, the activity during NREM sleep was significantly lower prior to the transition to REM sleep than to wakefulness (Fig. 4f). Together, these results confirmed our findings in the optrode recording experiments (Fig. 3), indicating that the great majority of vlPAG GABAergic neurons are REM-off, and the suppression of their activity is closely associated with REM sleep initiation and maintenance.

**Slow modulation of vlPAG activity during inter-REM interval.** REM sleep in rodents occurs every 10–15 min on average in the light phase, following an ultradian rhythm of REM/NREM

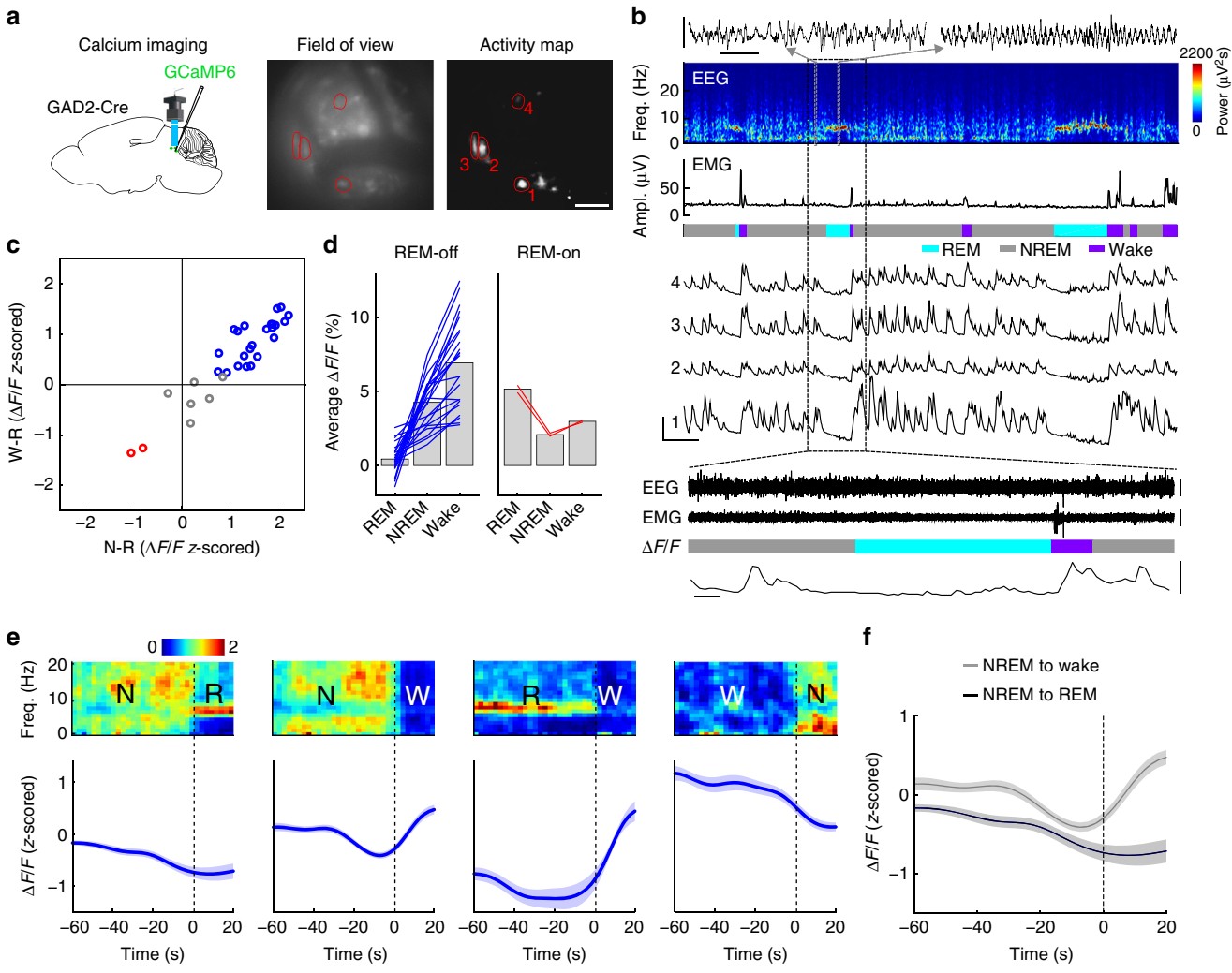

**Fig. 4** Sleep–wake activity of vlPAG GABAergic neurons measured with calcium imaging. **a** Left, schematic of calcium imaging through GRIN lens. Middle, field of view of an example imaging session. Right, pixel-wise activity map of an example imaging session. Red line, four example ROIs. Scale bar, 100 μm. Brain figure adapted from Allen Mouse Brain Atlas (© 2015 Allen Institute for Brain Science. Allen Brain Atlas API. Available from: http://brain-map.org/api/index.html). **b** EEG power spectrogram, EMG trace, color-coded brain state, and ΔF/F traces of the ROIs outlined in **a** (scale bars, 120 s, 20 % ΔF/F). Two example EEG raw traces (indicated by gray boxes) are shown on top of the EEG spectrogram (scale bars, 1 s, 0.3 mV). The black box indicates a region in which raw EEG, EMG (scale bars, 0.5 mV) and the fluorescence signal of ROI 2 are shown on an expanded timescale (scale bars, 10 s, 20 % ΔF/F). **c** Activity modulation of 31 ROIs from 3 mice. W wake, R REM, N NREM. Blue, significant REM-off ROIs ($P < 0.05$, Wilcoxon rank-sum test, post-hoc Bonferroni correction); red, significant REM-on ROIs; gray, other neurons. **d** Calcium activity of significant REM-off (left) and REM-on (red) ROIs during different brain states. Each line shows activity of one ROI; gray bar, average across ROIs. **e** Average EEG spectrogram (upper, normalized by the mean power in each frequency band) and mean calcium activity (ΔF/F, z-scored) of significant REM-off ROIs (lower) at brain state transitions ($n = 23$). Shading, ± s.e.m. **f** The mean calcium activity during NREM episodes preceding wake was significantly higher than those preceding REM episodes ($P = 0.003$, $T(22) = -3.24$, paired $t$-test)

alternation[3]. Since the vlPAG GABAergic neurons powerfully gate REM sleep, we wondered whether they also play a role in controlling its ultradian timing. Thus, in addition to the rapid activity changes at brain state transitions, we also analyzed their slow modulation during the period between successive REM sleep episodes (inter-REM interval), which consists of both NREM and wake episodes.

First, we temporally compressed each inter-REM interval to unit duration before averaging the vlPAG activity over multiple inter-REM intervals and across multiple REM-off GABAergic neurons. Both optrode recording and calcium imaging revealed a high activity at the beginning of the interval, which decreased slowly but consistently throughout the interval (Fig. 5a, Supplementary Fig. 10 and Supplementary Fig. 11a). Interestingly, when

we analyzed the activity within NREM and wake states separately during the inter-REM interval, the decrease was significant for NREM ($R = -0.48$, $P = 2.5 \times 10^{-4}$, $T(53) = -3.93$, linear regression) but not for wake states (Fig. 5b, Supplementary Fig. 11b; $R = -0.04$, $P = 0.76$, $T(52) = -0.29$). Consistent with this observation, the firing rate was significantly higher in the first than the last NREM episode within each inter-REM interval (Fig. 5c; $P = 0.008$, $T(10) = 3.28$, paired $t$-test; Supplementary Fig. 11c), but no significant difference was found for the wake episodes (Fig. 5d, $P = 0.15$, $T(10) = 1.56$; Supplementary Fig. 11d). Furthermore, when the activity was averaged across all NREM or wake episodes, the decrease was apparent within a single NREM but not wake episode (Fig. 5e; NREM, $R = -0.65$, $P = 2.8 \times 10^{-15}$, $T(108) = -8.77$; wake, $R = 0.21$, $P = 0.027$, $T(108) = 2.24$; Supplementary

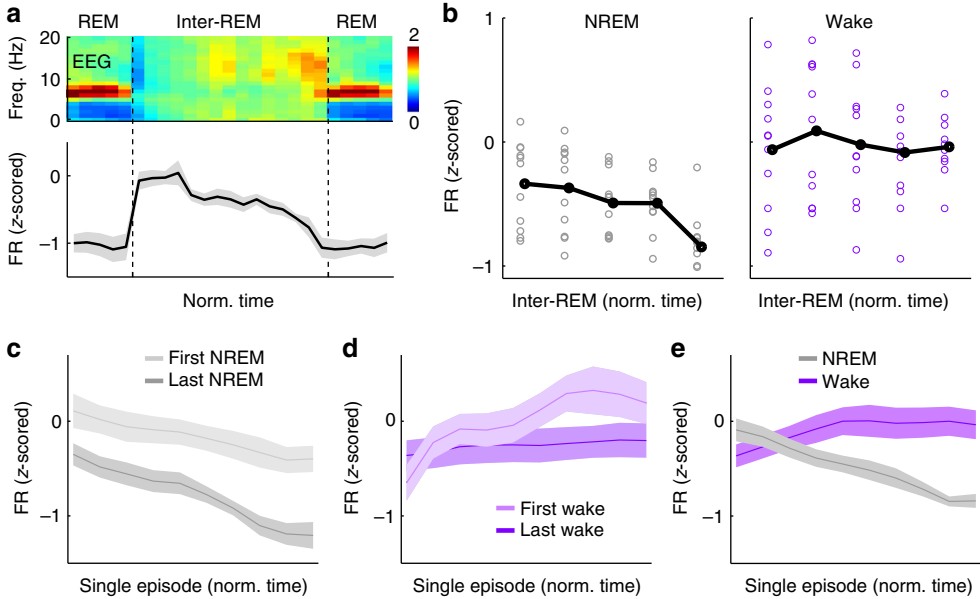

**Fig. 5** Slow modulation of vlPAG GABAergic neuron activity during inter-REM interval. **a** Average normalized EEG spectrogram (upper) and mean firing rate (z-scored) of significant REM-off vlPAG GABAergic neurons (lower) during two successive REM episodes and the inter-REM interval. Each REM episode and inter-REM interval was temporally compressed to unit length before the firing rates were averaged over multiple episodes/intervals and across GABAergic REM-off neurons (n = 11). Shading, ±s.e.m. **b** Firing rate (FR, z-scored) during NREM (left) and wake (right) episodes within different segments of the inter-REM interval. Each inter-REM interval was divided into five equally sized bins, and NREM or wake firing rates were averaged within each bin. Each symbol represents the average NREM or wake firing rate of a unit. The average NREM firing rate decreased during the inter-REM period (R = −0.48, P = 2.5 × 10⁻⁴, T(53) = −3.93, linear regression), while the wake activity showed no significant trend (R = −0.04, P = 0.76, T(52) = −0.29). Black line, average firing rate of each bin. **c** Mean firing rates during the first (light gray) and last (dark gray) NREM episodes of each inter-REM interval. Each NREM episode was temporally compressed to unit duration before the z-scored firing rate was averaged over episodes and across cells. The firing rate during the first NREM period was significantly higher than during the last period (n = 11 units, P = 0.008, T(10) = 3.28, paired t-test). Shading, ±s.e.m. **d** Mean firing rates during the first (light purple) and last (dark purple) wake episodes of each inter-REM interval. Each wake episode was temporally compressed to unit duration before averaging. **e** Mean firing rates during all NREM (gray) and wake (purple) episodes. Note that the firing rate decreased during NREM (R = −0.65, P = 2.8 × 10⁻¹⁴, T(108) = −8.78) but increased during wake episodes (R = 0.21, P = 0.027, T(108) = 2.24)

Fig. 11e). These findings suggest that the slow process modulating vlPAG activity during each inter-REM interval operates selectively during NREM sleep. Given the strong REM suppression effect of vlPAG GABAergic neurons (Fig. 1), the slow decrease of their firing rates could gradually enhance the propensity of the next REM episode and thus regulate the ultradian timing of the REM/NREM alternation. In contrast, the REM-on neurons in the unidentified population showed no consistent change in their NREM or wake activity during the inter-REM interval (Supplementary Fig. 12a–e; NREM, R = 0.13, P = 0.33, T(58) = 0.99; wake, R = −0.17, P = 0.19, T(58) = −1.32; linear regression).

**Homeostatic regulation of REM sleep and vlPAG activity.** The duration of the inter-REM interval is highly variable in rodents, and in the rat it was found to be correlated with the duration of the preceding REM episode[32, 33]. Here we found a similar relationship in the mouse (Fig. 6a; R = 0.39, P = 6.8 × 10⁻³⁶, T(970) = 13.03, linear regression), with longer inter-REM intervals following longer REM periods (Fig. 6b). Such a correlation suggests that the inter-REM interval is under homeostatic regulation, in which REM sleep pressure accumulated during inter-REM intervals is partially dissipated by each REM sleep episode[33, 34]; a longer episode causes stronger dissipation of the pressure and thus requires a longer interval for accumulation before the next REM episode. As a further test of this homeostatic regulation, we performed closed-loop activation of vlPAG GABAergic neurons (see Methods section), in which laser stimulation was applied as soon as spontaneous REM sleep onset was detected and lasted until the REM episode ended (in 50% of randomly selected REM

episodes). We found that shortening of REM episode duration by the stimulation (P = 0.002, T(5) = 5.88, paired t-test) also led to a significant shortening of the subsequent inter-REM interval, likely due to a reduced dissipation of REM sleep pressure (Fig. 6c; P = 0.003, T(5) = −5.29, paired t-test).

We then examined the effect of each REM episode on the NREM activity of vlPAG GABAergic neurons. Compared to the pre-REM activity, the NREM activity following a REM episode was significantly elevated (Fig. 6d; P = 0.006, T(10) = −3.43, paired t-test; Supplementary Fig. 13a), suggesting that a single REM episode can reset the firing rate of vlPAG GABAergic neurons, which has been decreasing slowly during the preceding inter-REM interval. In contrast, we found no significant difference in NREM activity before and after a single wake episode (Fig. 6e; P = 0.41, T(10) = 0.84, paired t-test; Supplementary Fig. 13b), suggesting that the firing rate reset is specific to REM sleep. The post-REM activity of vlPAG GABAergic neurons was higher following long (>90 s) REM episodes than following short (≤90 s) REM episodes (Fig. 6f; P = 0.03, T(3) = −3.8, paired t-test; Supplementary Fig. 13c), and the mean firing rate during the inter-REM interval was significantly correlated with the preceding REM episode duration (Fig. 6g; P = 0.001, T(38) = 3.58, linear regression; Supplementary Fig. 13d). In contrast, the activity of unidentified REM-on neurons showed no dependence on the preceding REM duration (Supplementary Fig. 14). These results point to a strong parallel between the modulation of vlPAG GABAergic neuron activity and the accumulation and dissipation of REM sleep pressure. Finally, we found a strong correlation between the duration of the inter-REM interval and the vlPAG firing rate during the interval (Fig. 6h; P = 1.6 × 10⁻⁵, T(38) = 4.95;

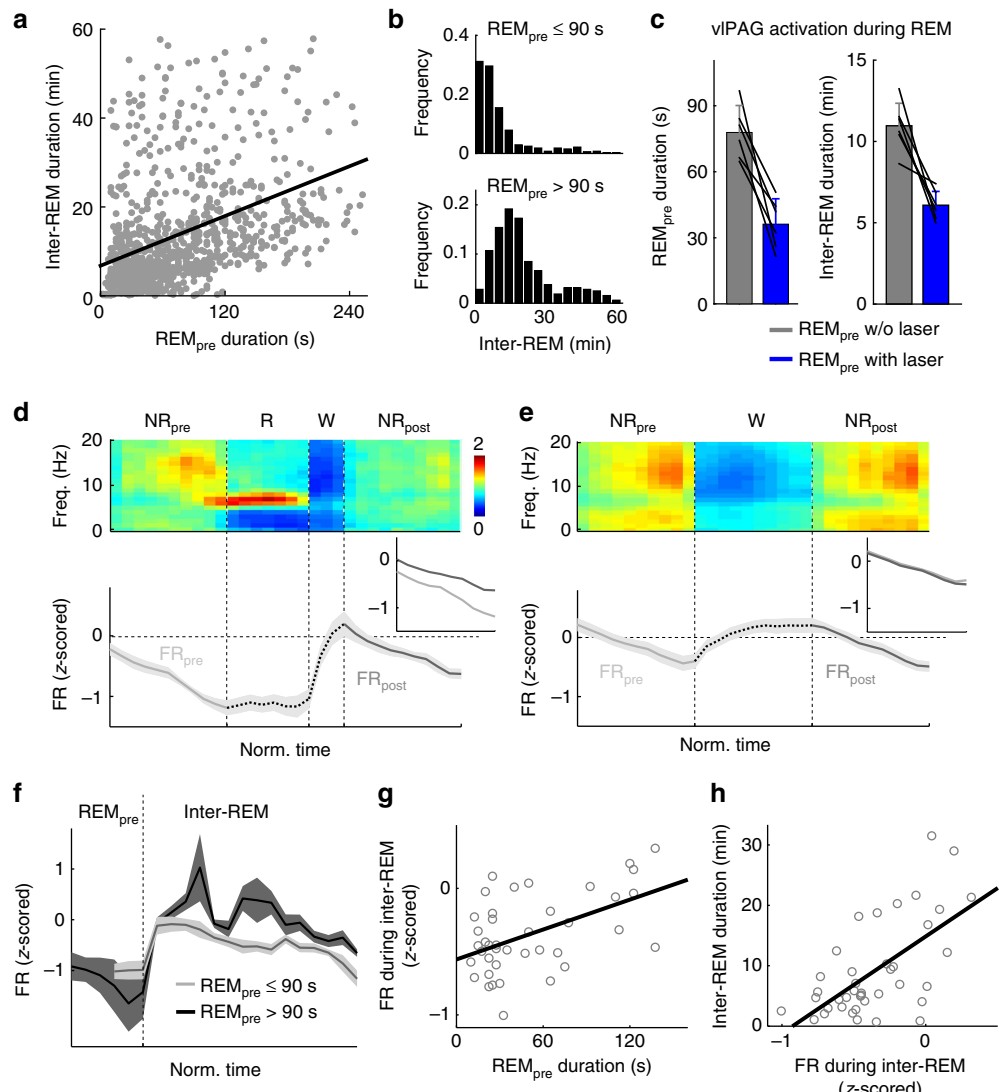

**Fig. 6** Homeostatic modulation of REM sleep and REM-off neuron activity. **a** Correlation between REM episode duration and subsequent inter-REM interval. Each dot represents a single episode ($n = 972$ episodes from 27 mice). Line, linear fit ($R = 0.39$, $P = 6.8 \times 10^{-36}$, $T(970) = 13.03$). **b** Distribution of inter-REM interval following short ($\leq 90$ s) and long ($> 90$ s) REM episodes. The two distributions are significantly different ($P = 2.8 \times 10^{-48}$, $z = -14.60$, Wilcoxon rank-sum test). **c** Effect of closed-loop activation of vlPAG GABAergic neurons on REM sleep duration and subsequent inter-REM interval. Closed-loop stimulation shortened both REM episodes ($n = 6$ mice, $P = 0.002$, $T(5) = 5.88$, paired $t$-test) and subsequent inter-REM intervals ($P = 0.003$, $T(5) = -5.29$). Lines, single mice. Error bar, ±s.d. **d** Average EEG spectrogram (upper) and mean firing rate ($z$-scored) of significant REM-off vlPAG GABAergic neurons (lower) during the NREM→REM→wake→NREM transition sequence. Each REM, wake, or NREM episode was temporally normalized. Shading, ±s.e.m. Inset, comparison of firing rates during the NREM episodes preceding ($NR_{pre}$) and following REM sleep ($NR_{post}$). The firing rate during $NR_{post}$ was higher than that during $NR_{pre}$ ($n = 11$ units; $P = 0.006$, $T(10) = -3.43$, paired $t$-test). **e** Similar to **d**, but for the NREM→wake→NREM sequence. Without the intervening REM episode, the firing rates were similar between $NR_{pre}$ and $NR_{post}$ ($P = 0.42$, $T(10) = 0.84$). **f** Comparison of vlPAG activity during inter-REM interval following short ($\leq 90$ s) and long ($> 90$ s) REM episodes. Following a long REM period, the firing rate was significantly higher than following a short one ($P = 0.03$, $T(3) = -3.78$, paired $t$-test). Shading, ±s.e.m. **g** Correlation between REM episode duration and vlPAG firing rate during the subsequent inter-REM interval. Each dot represents activity of a unit during a single inter-REM interval ($n = 40$). Line, linear fit ($R = 0.50$, $P = 0.001$, $T(38) = 3.58$). **h** Correlation between vlPAG firing rate during inter-REM interval and duration of the interval. Line, linear fit ($R = 0.63$, $P = 1.6 \times 10^{-5}$, $T(38) = 4.95$).

Supplementary Fig. 13e). This is consistent with the finding that higher vlPAG activity is associated with lower propensity for REM sleep (Figs. 1d–g, 3c, d and 4c, d), and it provides correlative evidence that the modulation of vlPAG activity by prior sleep history can mediate the homeostatic regulation of REM sleep.

### Discussion

Using optogenetic manipulation and Caspase 3-mediated cell ablation, we found that vlPAG GABAergic neurons powerfully suppress REM sleep. Activation of these neurons greatly reduced

both NREM→REM transitions and REM sleep maintenance while consolidating NREM sleep (Fig. 1), which are likely mediated in part by their projection to the dorsolateral pons (Fig. 2). Optrode recording and calcium imaging showed that their activity was strongly suppressed at the onset of each REM sleep episode and increased abruptly at its termination (Figs. 3 and 4), consistent with their functional role in gating REM sleep. In addition to the rapid changes at brain state transitions, their activity was also modulated on a timescale of minutes. The dependence of their firing rates on sleep history closely parallels the accumulation and

dissipation of REM sleep pressure (Figs. 5 and 6). These results support the notion that REM sleep onset and termination are controlled by fast synaptic inhibition from vlPAG GABAergic neurons to REM-promoting neurons in the pons, whereas the ultradian timing of REM sleep is regulated by a homeostatic process acting through slow modulation of vlPAG GABAergic activity.

The strong suppression of vlPAG activity during REM sleep (Figs. 3e and 4e) could be mediated in part by inhibitory inputs from REM-active neurons. In addition to the small number of REM-on neurons within the vlPAG[7] (Figs. 3c, d and 4c, d, red), which could locally inhibit the REM-off neurons, REM-active GABAergic neurons in the pons, ventral and dorsal medulla, lateral hypothalamus, and preoptic area have all been shown to project to the vlPAG[9, 11, 35]. In particular, rabies-mediated trans-synaptic retrograde tracing has shown that GABAergic neurons in the ventrolateral medulla, which powerfully promote REM sleep, directly innervate GABAergic vlPAG neurons[11]. Optogenetic activation of these axon projections promoted REM sleep at a magnitude similar to that of activating the medulla neuron cell bodies, suggesting that inhibition of the vlPAG REM-off neurons is a major mechanism promoting REM sleep.

Surprisingly, the REM-off vlPAG neurons were most active during wakefulness (Figs. 3 and 4), even though their optogenetic activation increased NREM sleep and decreased wakefulness (Figs. 1 and 2). One of the excitatory inputs driving their wake activity may be the projection from orexin/hypocretin neurons in the hypothalamus[9], which are most active during wakefulness[36], and vlPAG GABAergic neurons are known to express orexin/hypocretin receptors[10]. Given the power of vlPAG GABAergic neurons in suppressing REM sleep (Fig. 1), this input may effectively prevent unwanted intrusions of REM sleep into wakefulness. Although superficially the high wake activity of vlPAG GABAergic neurons may seem incompatible with the wake-suppressing effect of their optogenetic activation (Fig. 1), we note that the effect was primarily due to an enhanced NREM sleep maintenance. The low efficacy of these neurons in inducing wake→NREM transitions (Fig. 1f) may allow other wake-promoting neurons to counteract the high firing rates of the vlPAG neurons to maintain wakefulness. On the other hand, activation of these neurons was highly effective in reducing NREM→wake transitions (Fig. 1f); the increase in their natural firing rates at this transition indicates that natural awakening from NREM sleep is not due to, but in spite of, the firing rate change of vlPAG GABAergic neurons.

To account for the ultradian alternation between REM and NREM sleep, the first circuit model emphasized reciprocal interactions between cholinergic REM-on and monoaminergic REM-off neurons[37]. Subsequent studies have highlighted the importance of GABAergic inhibition between REM-on and REM-off neurons[7, 9, 11, 35]. Although fast GABAergic interactions can account for rapid transitions between brain states, they are insufficient to explain the temporal dynamics of NREM/REM alternation on a timescale of minutes (rodents) to hours (humans); a separate, slow-varying process is required[38, 39]. Here, in addition to confirming the correlation between the REM episode duration and subsequent inter-REM interval in mice (Fig. 6a), we found that shortening of REM sleep episodes through an optogenetic manipulation also shortened the subsequent inter-REM interval (Fig. 6c). These findings support the notion that the ultradian timing of REM sleep is strongly regulated by a homeostatic pressure that accumulates on a timescale of minutes and is partially dissipated by each REM sleep episode[33, 34, 39]. Interestingly, chemogenetic manipulations that shortened/prolonged REM sleep duration also decreased/increased the EEG delta power in subsequent NREM sleep[12],

suggesting that, in addition to dissipating REM sleep pressure, REM sleep also contributes to the accumulation of NREM sleep pressure.

At the neuronal level, we found that vlPAG GABAergic neuron activity decreased gradually during the inter-REM interval (Fig. 5) and was reset by each REM episode, the magnitude of which depended on the episode duration (Fig. 6). These firing rate modulations mirror the accumulation and dissipation of REM sleep pressure[33, 34], and implementation of such slow modulations of REM-off neurons in circuit models can account for the temporal dynamics of REM/NREM alternations[38, 39]. Previous studies have shown that REM sleep deprivation for several hours decreased/increased the activity of REM-off/REM-on neurons[40] and increased the expression of brain-derived neurotrophic factor in the pons[41, 42]. It would be interesting to know whether these processes also operate on a timescale of minutes and whether they are causally linked to the firing rate modulations in vlPAG neurons (Figs. 5 and 6). Regarding the cellular mechanisms for these slow activity modulations, recent work in flies showed that sleep pressure is reflected by the change in excitability of sleep-promoting neurons projecting into the dorsal fan-shaped body, regulated through the antagonistic actions of two types of potassium channels[43]. It would be interesting for future studies to test whether conductances of vlPAG GABAergic neurons are similarly regulated by homeostatic need for REM sleep in the mammalian brain.

## Methods

**Animals**. All experimental procedures were approved by the Animal Care and Use Committee at the University of California, Berkeley. Optogenetic manipulation experiments were performed in male or female *GAD2*-Cre (Jackson Laboratory stock 010702) and *VGLUT2*-Cre (016963) mice. Optrode recording, calcium imaging, lesion, and rabies-mediated trans-synaptic tracing experiments were all performed in male or female *GAD2*-Cre, *VGLUT2*-Cre, *TH-Cre* (008601), and *ChAT-Cre* (006410) mice. Animals were housed on a 12-h dark/12-h light cycle (light on between 0700 and 1900 hours). Animals with implants for EEG/EMG recordings, optogenetic stimulation, optrode recordings, or calcium imaging were housed individually. Mice injected with viruses for trans-synaptic tracing were housed in groups of up to six animals.

**Surgical procedures**. Adult (6–12-week-old) mice were anesthetized with isoflurane (3% induction, 1.5% maintenance) and placed on a stereotaxic frame. Body temperature was kept stable throughout the procedure by using a heating pad. After asepsis, the skin was incised to expose the skull, and the overlying connective tissue was removed. For access to the vlPAG, a craniotomy (1 mm diameter) was made on top of the right superior colliculus (4.6–4.8 mm posterior to bregma, 0.6–0.7 mm lateral). AAV2-EF1a-FLEX-ChR2–eYFP, AAV2-EF1a-FLEX-eYFP (produced by University of North Carolina Vector Core), AAV1-Syn-Flex-GCaMP6f (University of Pennsylvania Vector Core), or AAV-DJ-EF1a-FLEX-ChR2–eYFP (University of Stanford Gene Vector and Virus Core) was loaded into a sharp micropipette mounted on a Nanoject II or Nanoject III and injected slowly at a depth of 2.5 mm from the brain surface (300–500 nl, unilateral). An optical fiber (0.2 mm diameter) was inserted with the tip 0.2 mm above the virus injection site. To stimulate axon projections of vlPAG GABAergic neurons in the dorso-lateral pons, the optical fiber was implanted within the pons (5.2–5.3 mm posterior to bregma, 0.9 mm lateral, 3.0 mm depth). For cell-type-specific lesion experiments, AAV2-Flex-taCasp3-Tevp (University of North Carolina Vector Core) was injected bilaterally into the vlPAG (200–300 nl). For optogenetic experiments, data from animals where the tip of the optical fiber was not within the aimed location were excluded. For lesion experiments, data from animals where virus expression was not restricted to the vlPAG were excluded.

For EEG and EMG recordings, a reference screw was inserted into the skull on top of the left cerebellum. EEG recordings were made from two screws, one on top of hippocampus (about 2 mm posterior to bregma) and one on top of the prefrontal cortex (about 1 mm anterior to bregma). Two EMG electrodes were inserted into the left and right neck muscles. For EEG/EMG recordings in differential mode, four screws were fixed on the skull, two posterior and two anterior of bregma. All screws, electrodes, microdrives, connectors, and GRIN lenses required for EEG/EMG recordings, optrode recordings, and calcium imaging were secured to the skull using dental cement.

**Rabies virus tracing**. For retrograde tracing, AAV-CAG-FLEx$^{loxP}$-TC$^{66T}$ (1.0 × 10$^{12}$ gc/mL) and AAV8-CAG-FLEx$^{loxP}$-RG (1.8 × 10$^{12}$ gc/mL) was mixed at a 1:1

ratio and stereotaxically injected. For tracing from glutamatergic neurons in the dorsolateral pons, 100 nl was injected into *VGLUT2-Cre* mice at −5.3 mm AP, 0.8 mm ML, 3.3 mm DV, unilaterally. For tracing from the LDT, 300−500 nl was injected in *ChAT-Cre* mice at −5.2 mm AP, 0.7 mm ML, 3.2 mm DV, bilaterally. For tracing from the noradrenergic neurons in the LC, 200−300 nl was injected into *TH-Cre* mice at −5.34 mm AP, 0.8 mm ML, 3.2 mm DV, unilaterally. Two weeks after the AAV injection, 300−400 nl of RV*dG*-eGFP+EnvA ($1.0 \times 10^9$ cfu/ml) was injected into the same location. The injection pipette for AAV was tilted at 20 degrees from vertical while the injection pipette for RV was inserted vertically to limit virus coinfection along a common injection tract. We excluded animals with inefficient trans-synaptic labeling indicated by <30 rabies-GFP-labeled cells detected outside of the injection site.

**Histology and immunohistochemistry**. Mice were deeply anesthetized and transcardially perfused with 0.1 M phosphate-buffered saline (PBS) followed by 4% paraformaldehyde in PBS. After removal, brains stayed overnight in 4% paraformaldehyde. For cryoprotection, brains were stored in 30% sucrose (w/v) in PBS solution for at least 1 night. Brains were sliced in 30 or 40 µm sections using a cryostat (Leica). For immunohistochemistry, non-specific binding sites were blocked by incubating the brain sections in 2% goat serum (Millipore) in PBST (0.3% Triton X-100 in PBS).

To amplify the fluorescence of axon fibers expressing ChR2-eYFP or eYFP, we applied antibodies for GFP (A11122, Life Technologies, 1:1000). Brain sections were incubated with the primary antibody diluted in blocking solution for 2 nights. A species-specific secondary antibody conjugated with green Alexa fluorophore (A11008, Life Technologies, 1:1000; goat anti-rabbit) was diluted in PBS and applied for 2 h. Fluorescence images were taken using a confocal microscope (LSM 710 AxioObserver Inverted 34-Channel Confocal, Zeiss), fluorescence microscope (Keyence, BZ-X710), and Nanozoomer 2.0 RS (Hamamatsu).

**Fluorescence in situ hybridization**. FISH was performed to detect *GAD2* mRNA. Brains were embedded and mounted with Tissue-Tek OCT compound (Sakura finetek), and 30 µm sections were cut using a cryostat (Leica). FISH was performed using an RNAscope assay according to the manufacturer's instructions (RNAscope® Fluorescent Multiplex Reagent Kit, Cat. # 320850, Advanced cell Diagnostics). Sections were hybridized with a *GAD2* probe (Mm-*Gad2*, 439371) and eGFP probe (EGFP-C3, 400281-C3), and amplification steps were carried out followed by Hoechst staining. Fluorescence images were taken using a confocal microscope (LSM 710 AxioObserver Inverted 34-Channel Confocal, Zeiss) or a fluorescence microscope (BZ-X700, Keyence).

**Polysomnographic recordings**. EEG and EMG electrodes were connected to flexible recording cables via a mini-connector, and recordings were made in the animal's home cage placed in a sound-attenuated box. Recordings started after at least 1 h of habituation. The signals were recorded with a TDT RZ5 amplifier (bandpass filter, 1−750 Hz; sampling rate, 1500 Hz) or an A-M Systems amplifier (sampling rate, 600 Hz). For the recordings using the TDT amplifier, EEG and EMG signals were referenced to a common ground screw, placed on top of the cerebellum. Recordings with the A-M Systems amplifier were made in differential mode. We used the difference between the voltage potentials recorded from an anterior (on top of prefrontal cortex) and posterior screw (on top of hippocampus) as EEG signal and the difference between the potentials from the two EMG electrodes as EMG signal.

To determine the brain state from the recorded EEG and EMG signals, we first calculated the power spectrum for the EEG and EMG signals using 5 s sliding windows, sequentially shifted by 2.5 s increments. Next, we summed the EEG power in the range from 1 to 4 Hz and from 6 to 12 Hz, yielding time-dependent delta and theta power. For further analysis, we computed the theta/delta power ratio. We also summed the EMG power in the range 20−300 Hz. As a first step, we determined for each time point the brain state using an automated threshold algorithm. We first determined a threshold for the delta power (delta threshold), separating the typically bimodal distribution of the delta power into a lower and higher range. A state was assigned as NREM sleep if the delta power was larger than the delta threshold and if the EMG power was lower than its mean plus one standard deviation. A state was classified as REM if (1) the delta power was lower than the delta threshold, (2) the theta/delta ratio was higher than one standard deviation above the mean, and (3) the EMG power was lower than its mean plus one standard deviation. All remaining states were classified as wake. The wake state thus encompassed states with high EMG power (active awake) or low delta power but without elevated EMG activity or theta/delta ratio (quiet awake). Finally, we manually verified the automatic classification using a custom-built graphical user interface programmed in Matlab.

**Optogenetic stimulation**. We performed optogenetic activation experiments 4−6 weeks after injection of AAV-expressing ChR2. Experiments were performed in the afternoon (1200 and 1900 hours) and lasted at least 6 h. For optogenetic stimulation of GABAergic vlPAG neurons, each trial consisted of a 20 Hz pulse train lasting for 300 s using a blue 473-nm laser (6−8 mW at fiber tip, Shanghai

Laser). The inter-trial interval was randomly chosen from a uniform distribution from 15 to 25 min.

To test the role of vlPAG GABAergic neurons in REM sleep maintenance, we applied a closed-loop stimulation protocol. Sleep/wake states were classified based on EEG and EMG in real time. When REM sleep was detected, the laser was turned on with 50% probability and turned off only when the REM episode ended. This allowed comparison of the REM episode durations and the subsequent inter-REM intervals with and without laser stimulation within the same recording session.

**Transition analysis**. To quantify transition probabilities between brain states, we discretized time into 20 s bins and aligned all laser stimulation trials from all *N* mice by the onset of laser stimulation at time 0.

To determine the transition probability from state *X* to *Y* for time bin *i*, $P_i(Y|X)$, we first determined the number of trials (*n*) in which the animal was in brain state *X* during the preceding time bin *i−1*. Next, we identified the subset of these trials (*m*) in which the animal transitioned into state *Y* in the current time bin *i*. The transition probability $P_i(Y|X)$ was computed as *m/n*. In Fig. 1d, each bar represents the transition probability averaged across three consecutive 20 s bins. To compute the baseline transition probabilities, we averaged across all time bins excluding the laser stimulation period and the 2 min period following laser stimulation.

**Ablation of vlPAG GABAergic neurons**. To test the impact of ablating vlPAG GABAergic neurons, we recorded the EEG and EMG of mice expressing pro-Caspase 3 in the vlPAG, 2−3 weeks after virus injection. Each mouse was habituated for at least 1 day, and sleep−wake behavior during the light or dark cycle was recorded for at least 2 days. For control, we recorded from mice expressing eYFP that underwent identical procedures.

**Optrode recording**. To record single unit activity from vlPAG GABAergic neurons, we used custom-built optrodes, consisting of 12−14 microelectrodes (Stablohm 675, California Wire Company) twisted into stereotrodes or tetrodes and attached to an optic fiber (0.2 mm diameter). The electrodes were inserted into a screw-driven microdrive and soldered to a $2 \times 10$ connector. For EEG/EMG recordings, two additional wires were attached to the connector. For stability, all components were attached to a custom-designed circuit board (fabricated by oshpark.com). The tips of the electrodes were electro-plated in a gold solution (Sifco 5355) mixed with 0.1% PEG solution (Sigma). Each electrode was plated iteratively, till reaching an impedance of 500−800 kΩ. Neurophysiological signals were recorded using a TDT RZ5 amplifier (Tucker-Davis Technologies) at a sampling rate of 25 kHz and bandpass filtered (0.3−8 kHz). The optrode was slowly lowered in 50 µm steps to search for well-separated, light-responsive units. Recordings were performed during the light cycle and lasted for at least 60 min. For each mouse, recordings were carried out over a period of 1−2 months, starting 3 weeks after virus injection.

Before sacrificing the animal, an electrolytic lesion was made to mark the end of the electrode tract by passing a current (100 µA, 10 s) through two electrodes. Based on the location of the lesion and the optrode tract, the anatomical location of each unit was reconstructed. We excluded units whose anatomical location was outside the vlPAG.

**Spike sorting**. Spikes were sorted offline based on the waveform energy, the three largest principal components for each spike waveform on each stereotrode or tetrode channel, and a further parameter measuring for each potential unit the distance between spontaneous and laser evoked waveforms (as quantified by the inner product). Single units were identified either manually using the software Klusters (http://neurosuite.sourceforge.net) or automatically using the software KlustaKwik (http://klustakwik.sourceforge.net). Only units with a clear refractory period in the auto-correlogram and satisfying criteria for isolation distance (>20) and *L*-ratio (<0.15) were included in the data set. The median values of the isolation distance and L-ratio for all units were 36.9 and 0.022, respectively. Two units recorded with the same stereotrode or tetrode on different days were counted as the same unit if their waveforms were similar, their recording depths differed by <100 µm, and if their average firing rates were not significantly different.

**Optogenetic tagging**. To identify ChR2-expressing vlPAG GABAergic neurons, high-frequency laser pulse trains (15 and 30 Hz with durations of 1 and 0.5 s, respectively) were delivered every minute. In a subset of recordings, we presented a 0.1 or 0.2 s long step pulse instead of the 15 Hz pulse train. The recorded units were classified into three categories (identified GABAergic units, laser-inhibited, or laser-unmodulated units) using the following approach: First, we tested whether the activity of each unit was significantly modulated by single laser pulses using a previously developed statistical test (SALT)[44]. This method probes whether the distribution of spike patterns (within 30 ms windows) elicited by single laser pulses is different from spontaneous spike patterns. Second, to determine whether a significantly modulated unit was excited or inhibited by laser stimulation, we tested whether the average firing rate during the 30 Hz laser pulse train was significantly increased or decreased relative to the firing rate in the 0.5 s window preceding laser stimulation ($P < 0.05$, Wilcoxon signed-rank test). Finally, among those that are excited by the laser stimulation, we identified units that followed each laser pulse

with short first-spike latency (<10 ms) and high reliability (>0.4) as GABAergic units. We excluded units exhibiting longer latencies or lower reliabilities. Units that were inhibited or not significantly modulated by laser stimulation were summarized as unidentified.

In total, we recorded 19 identified and 22 unidentified neurons. The low number of unidentified neurons is likely due to a sampling bias in our recording strategy: As the success rate for finding a driven unit in the vlPAG is generally low, we spent the 1–2 h on recording only if we found at least one unit that appeared to be activated by the laser pulses. As a result, most of the unidentified units were recorded simultaneously with other laser-driven, putative GABAergic units.

**Behavioral monitoring**. To identify the awake behaviors during which vlPAG GABAergic units are activated, we made video recordings (sampling rate of 5 Hz) using a camera placed on top of the mouse cage along with the electrophysiological signals. Active behaviors were divided into five categories: eating, grooming, moving, running, and quiet wake. The behavioral scoring was performed manually using a custom written graphical user interface (programmed in MATLAB).

**Firing rate calculation**. After spike sorting, the activity of each unit was represented as a spike train (sequences of 0 s and 1 s) with 0.67 ms time resolution. For firing rate calculation, spike trains were downsampled to 2.5 s time bins aligned with the brain state annotation. To reduce noise, spike trains were filtered with an exponential kernel with a time constant of 7.5 s. To quantify the firing rate dynamics during each sleep or wake episode, we normalized the length of each episode by dividing it into a fixed number of bins, allowing us to average across episodes and neurons.

**Prediction of brain state transitions**. To test whether the NREM activity of vlPAG GABAergic neurons is predictive of whether the animal will transition to REM sleep or wakefulness, we performed a classification analysis. Using Fisher's linear discriminant, we tested how well the future brain state (REM or wake) can be predicted from the neural activity in each 10 s bin from 0 to 60 s before the actual transition. The data set used for classification included the neural activities recorded up to 60 s before the transition. As the trials with NREM→Wake transitions outnumbered NREM→REM trials, we balanced the data set to avoid systematic prediction biases. The resulting data set included all NREM→REM trials and an equal number of randomly selected NREM→wake trials. To quantify the prediction accuracy, we performed a 10-fold cross-validation: The data set was randomly divided into 10 training and test sets, each comprising 90 and 10% of the trials, respectively. The classifier was trained on the training sets (comprising all time bins preceding the transition), and its performance was evaluated at each time bin using the test sets. The resulting accuracy value quantified how well, in average, the future transition can be predicted from the activity of a single neuron. Confidence intervals and P-values were calculated using bootstrapping by repeating the cross-validation procedure 100 times.

**Calcium imaging**. Calcium imaging experiments were performed 3–8 weeks following injection of AAV-expressing GCaMP6. Imaging sessions took place during the light cycle in the home cage placed within a sound-attenuated chamber. The animal was briefly anesthetized with isoflurane to secure the microscope to the baseplate and to focus it to a given field of view. The animal was then allowed to recover from anesthesia and habituate in the home cage for at least 30 min prior to imaging. Calcium activity was acquired using the nVista hardware and nVista HD software (Inscopix), with a 10 Hz image acquisition rate using 0.2–0.7 mW illumination. An Intan RHA2000 system (Intan Technologies) was used to simultaneously record EEG and EMG activity (bandpass filter, 1–750 Hz; sampling rate, 25 kHz). An output signal (20 Hz) delivered from the Inscopix system to the Intan system throughout the recording session was used to synchronize the timing between the imaging and EEG/EMG recordings. Recording sessions lasted between 40 and 110 min and were repeated up to 6 times per animal, spanning across 5 different days.

**Calcium imaging analysis**. Imaging data were processed in Python and Matlab (Mathworks). First, the acquired images were spatially downsampled by a factor of 4. To correct for lateral motion of the brain relative to the GRIN lens, we applied an algorithm based on a previous study[45]. We created a high-pass filtered image stack by calculating the difference between the image stack and a spatially low-pass filtered version of the image stack. A high-contrast region of the mean projection of the high-pass filtered image stack was selected as a reference. The spatial shift that maximized the cross-correlation between the reference region and each frame of the high-pass filtered image stack was calculated to obtain the motion corrected images.

Regions of interest (ROIs) were selected using a graphical user interface (programmed in Python) by manually identifying cell-body-sized regions with fluorescence activity across time and defining the coordinates of a polygon that contains the pixels of the ROI. The pixel intensities within each ROI were averaged to create a fluorescence time-series. In cases where the same ROI was identified across imaging sessions, the ROI was grouped together for analysis.

To correct for contamination of the fluorescence signal by out-of-focus neuropil, we calculated the neighboring neuropil signal, $F_{neuropil}$, from a bordering concentric ring with 20 μm distance from the perimeter of each ROI, excluding other ROIs, and subtracted $F_{neuropil}$, scaled by a correction factor, cf, from the raw ROI signal, $F_{raw}$ ($F_{neuropil-subracted} = F_{raw} - cf \times F_{neuropil}$). The cf for each recording session was estimated by calculating the ratio between the mean pixel intensity of a manually selected blood vessel, $F_{blood\ vessel}$, and a nearby region lacking an ROI signal, $F_{near\ blood\ vessel(no\ ROI)}$, each subtracted by the mean pixel intensity of an off-lens region, $F_{off-lens}$ (cf = [$F_{blood\ vessel} - F_{off-lens}$]/[$F_{near\ blood\ vessel(no\ ROI)} - F_{off-lens}$]). In cases where a blood vessel was not clearly present, the mean value obtained from other animals, cf = 0.55, was used.

The baseline of each neuropil-subtracted fluorescence time-series was estimated by calculating the best linear fit using periods of low fluorescence activity, represented by the 20th percentile of each recording session. Slope values were typically smaller than zero and thus allowed compensating for a drift in the fluorescence signal due to photobleaching of GCaMP.

**Statistics**. For optogenetic and ablation experiments, *GAD2-Cre* mice were randomly assigned to control (injected with AAV-expressing eYFP) and experimental groups (injected with AAV-expressing ChR2-eYFP or Casp3). No randomization was used for rabies-mediated trans-synaptic tracing. Investigators were not blinded to animal identity and outcome assessment.

All statistical tests (paired *t*-test, Wilcoxon rank-sum test, Wilcoxon signed-rank test, bootstrap) were two-sided. For both paired and unpaired tests, we ensured that the variances of the data were similar between the compared groups. For *t*-tests, we verified that the data were normally distributed using Lilliefors test for normality.

The 95% confidence intervals (CIs) for brain state percentages and transition probabilities (Figs. 1c, d and 2b) were calculated using a bootstrap procedure. For an experimental group of *n* mice, we calculated the CI as follows: we repeatedly resampled the data by randomly drawing for each mouse *p* trials (random sampling with replacement). For each of the *m* iterations, we recalculated the mean brain state percentages or transition probabilities across the *n* mice. The lower and upper CIs were then extracted from the distribution of the *m* resampled mean values. To test whether a given brain state or transition probability was significantly modulated by laser stimulation, we calculated for each bootstrap iteration the difference between the averaged brain state percentages or transition probabilities during laser stimulation and the corresponding baseline values for periods without laser stimulation. From the resulting distribution of difference values, we then calculated a P-value to assess whether laser stimulation significantly modulated brain states or transitions between brain states.

**Sample sizes**. For optogenetic activation experiments, cell-type-specific ablation experiments, and in vivo recordings (optrode recordings and calcium imaging), we continuously increased the number of animals until statistical significance was reached to support our conclusions. For rabies-mediated and anterograde tracing experiments, the selection of the sample size was based on numbers reported in previous studies. For optrode recordings, we first recorded a preliminary data set of six units from two mice. Based on analysis of this data set and given the success rate in finding identified GABAergic units, we predicted that about 20 units are sufficient to statistically support our conclusions.

**Code availability**. The code used in this study will be made available upon reasonable request.

**Data availability**. The data that support the findings of this study will be made available upon reasonable request.

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

## Acknowledgements

We thank Zhe Zhang, Nikolai Hörmann, and Andrei Popescu for technical help. We are grateful to Liqun Luo for providing viral reagents for trans-synaptic tracing. This work was supported by a Human Frontier Science Program (HFSP) postdoctoral fellowship (to F.W.).

## Author contributions

F.W., J.P.H.D., and Y.D. conceived and designed the experiments. F.W. and J.P.H.D. performed optogenetic activation experiments. F.W. performed optrode recordings and pro-Caspase 3-mediated lesion experiments. J.P.H.D. performed calcium imaging experiments and trans-synaptic viral tracing. S.C. performed in situ hybridization and fluorescence microscopy. M.B. and M.S.D. performed histology, immunohistochemistry experiments, and fluorescence microscopy. K.T.B. provided viral reagents for rabies-mediated trans-synaptic experiments. F.W. and J.P.H.D. analysed the data. F.W., J.P.H.D., and Y.D. wrote the manuscript.

## Additional information

**Competing interests:** The authors declare no competing financial interests.

