## [Peer Review File · Nature Communications]

Editorial Note: This manuscript has been previously reviewed at another journal that is not operating a transparent peer review scheme. This document only contains reviewer comments and rebuttal letters for versions considered at Nature Communications. Reviewer #2 from the earlier round of review was not available to comment during the review process at Nature Communications. As such a new Reviewer #3 was added to the review process.

Reviewers' comments:

Reviewer #1 (Remarks to the Author):

I had previously reviewed this paper and appreciate the authors' efforts to address my concerns. My prior and major criticisms of this paper is that it provided only an incremental advance in our understanding of REM sleep control and that it is largely confirmatory, if not in a part a replication, of prior work, including from the authors' laboratories (e.g., Weber et al., Nature, 2016). Given that very little has changed in this regard in the revision, my enthusiasm for this paper has not been substantially altered. For me, the compelling intellectual advance in this paper is the finding that the activity of vPAG GAD2 cells decreases predictably and progressively during the inter-REM interval, suggesting a mechanism underlying the ultradian rhythm of REM/NREM sleep alternations. The authors in fact claim that this is the "main focus" of their study, yet this data feels somewhat buried in the narrative and it remains only a correlative finding. Not to put too fine of a point on this issue, but even the title of the paper fails to accurately communicate this truly novel aspect of the paper. That said, this is a good paper, but could be a great paper if the authors were to drill down on the mechanism of the ultradian rhythm, showing for example necessity and sufficiency of this putative mechanism in REM homeostasis, and the neurobehavioral consequences of disrupting this process. In its present form, the paper repeatedly promises mechanisms, but instead delivers predictions based upon modeling, most acutely when it comes to the ultradian 'mechanism'.

Minor comments:

As noted by Reviewer #2, there is no convincing data in REF #22 to support a role for cholinergic LDT/PPT in REM sleep induction. At minimum, the authors should consider citing Kroeger et al., 2017 who came to a different conclusion, albeit using a different technical approach.

The authors use the term "pharmacogenetic" throughout their papers when describing DREADD-based work. I would strongly encourage replacing this term with "chemogenetic". To the reviewer's understanding, the term "pharmaco-genetic" was coined by Dr. Bryan Roth, the inventor of DREADDs. Because this term was meant, in part, to convey that this new methodology was in parallel with optogenetic methods, this was quite reasonable. The use of this term was however problematic from the standpoint that the same term was being used by molecular pharmacologists to describe research on genetic mutations that change the response to drugs at an organismal level (i.e., "personalized medicine"). And so most in the field, including Bryan Roth, using the evolved DREADD systems instead use the term "chemogenetic" in describing this methodology.

Reviewer #3 (Remarks to the Author):

In this paper Weber et al. investigates the role of vPAG in the control of REM sleep. They show that vPAG GABAergic neurons are participating both in initiation and maintenance of REM. The authors use both optogenetic tagging and calcium imaging to track the activity of vPAG GABAergic cells and the two types of measurements show a surprising degree of similarity. By quantifying slow firing rate changes during REM and NREM periods the authors show that neural activity correlates with REM pressure.

Although the results are not entirely surprising, this appears to be a solid work presenting valuable data about REM control. The level of agreement between recording and imaging is striking. The authors go a step beyond regular data analysis in sleep studies typically only calculating REM and

NREM firing rates by analyzing slow rate slopes within the segments. The effect of closed loop stimulation on the duration of post-REM sleep segments is particularly interesting.

The authors did a thorough job of answering the previous reviews. I welcome better anatomical characterization and more raw data (should appear in supplemental figures, see below). The prediction of future state transitions from firing rates is a valuable addition.

Major points:

1. From the bar graphs it looks like vIPAG neurons can reach 20 Hz firing rates during extended periods of time. Therefore, 20 Hz stimulation does not necessarily increase the activity of GABAergic cells, since probably through local inhibition non-light-evoked spikes are absent during stimulation (Fig.3a). Is there any proof that 20 Hz can be indeed interpreted as 'activation'? What would be the effect of say 50 Hz stimulation?

2. The number of recorded neurons is low. Optogenetic tagging can indeed result in low yield, but I am surprised at the very low number of non-tagged neurons at the same time, recorded from 6 (!) mice. Is there an explanation? Was this small sample balanced across mice? (How many tagged cells for each mouse?) Nevertheless, I find the combined number of recorded and imaged GABAergic neurons sufficient for the claims.

3. From Fig.S1 it appears some optic fiber implants were actually below the injected site. Was there an effect on sleep states in these mice as well? I am a bit surprised that unilateral inhibition was sufficient to generate strong effects. Therefore my broader question is how consistent these optogenetic effects were across animals? The authors only show averages.

Minor points:

1. p.8. "unidentified cell population contained a lower percentage of REM-off neurons" - although one can count the points in the figures, please quantify.

2. The authors did a thorough job answering the critics of reviewers. In this regard I welcome the addition of more data with less processing; nevertheless, raster plots of individual example cells would still increase my enthusiasm about the data. I also suggest incorporating some of the non-normalized firing rate data from the reviewer figures into the supplementary material.

Reviewer #1

I had previously reviewed this paper and appreciate the authors' efforts to address my concerns. My prior and major criticisms of this paper is that it provided only an incremental advance in our understanding of REM sleep control and that it is largely confirmatory, if not in a part a replication, of prior work, including from the authors' laboratories (e.g., Weber et al., Nature, 2016). Given that very little has changed in this regard in the revision, my enthusiasm for this paper has not been substantially altered. For me, the compelling intellectual advance in this paper is the finding that the activity of vIPAG GAD2 cells decreases predictably and progressively during the inter-REM interval, suggesting a mechanism underlying the ultradian rhythm of REM/NREM sleep alternations. The authors in fact claim that this is the "main focus" of their study, yet this data feels somewhat buried in the narrative and it remains only a correlative finding. Not to put too fine of a point on this issue, but even the title of the paper fails to accurately communicate this truly novel aspect of the paper. That said, this is a good paper, but could be a great paper if the authors were to drill down on the mechanism of the ultradian rhythm, showing for example necessity and sufficiency of this putative mechanism in REM homeostasis, and the neurobehavioral consequences of disrupting this process. In its present form, the paper repeatedly promises mechanisms, but instead delivers predictions based upon modeling, most acutely when it comes to the ultradian 'mechanism'.

In this study, we first confirmed the role of vIPAG GABAergic neurons as REM-off neurons, provided a detailed analysis of the effects of their activation on brain state transitions, and demonstrated their monosynaptic innervation of several wake- and sleep-regulatory cell types. We then went beyond the existing model and provided evidence for a role of these neurons in controlling the mammalian REM-NREM sleep cycle through their slow firing rate modulations. We agree that the latter finding is correlative at this stage. Proving necessity and sufficiency of this mechanism for REM sleep homeostasis would require a tool to selectively disrupt the slow firing rate modulation without affecting the overall firing rate, which is unfortunately not available. Understanding the physiological basis of the ultradian sleep cycle requires knowing which neurons are involved, the relationship between their activity and brain states, and what neural or molecular mechanisms regulate their activity. Our study demonstrates a strong relationship between vIPAG GABAergic neuron activity and REM sleep pressure, which is an important step in understanding the mechanism for REM sleep homeostasis. Although correlative, it provides an important entry point for future studies to investigate the biological processes regulating the firing rates of these neurons and REM sleep pressure.

Minor comments:

As noted by Reviewer #2, there is no convincing data in REF #22 to support a role for cholinergic LDT/PPT in REM sleep induction. At minimum, the authors should consider citing Kroeger et al., 2017 who came to a different conclusion, albeit using a different technical approach.

We thank the reviewer for the great suggestion and have now cited Kroeger et al. (2017) and discussed their different finding (p. 7, lines 22–23).

The authors use the term “pharmacogenetic” throughout their papers when describing DREADD-based work. I would strongly encourage replacing this term with “chemogenetic”. To the reviewer’s understanding, the term “pharmaco-genetic” was coined by Dr. Bryan Roth, the inventor of DREADDs. Because this term was meant, in part, to convey that this new methodology was in parallel with optogenetic methods, this was quite reasonable. The use of this term was however problematic from the standpoint that the same term was being used by molecular pharmacologists to describe research on genetic mutations that change the response to drugs at an organismal level (i.e., “personalized medicine”). And so most in the field, including Bryan Roth, using the evolved DREADD systems instead use the term “chemogenetic” in describing this methodology.

We thank the reviewer for this clarification and have replaced “pharmacogenetic” with “chemogenetic” throughout the manuscript.

Reviewer #3

Major concerns

1. From the bar graphs it looks like vIPAG neurons can reach 20 Hz firing rates during extended periods of time. Therefore, 20 Hz stimulation does not necessarily increase the activity of GABAergic cells, since probably through local inhibition non-light-evoked spikes are absent during stimulation (Fig.3a). Is there any proof that 20 Hz can be indeed interpreted as 'activation'? What would be the effect of say 50 Hz stimulation?

Figure R1. Pulsed laser stimulation increases the firing rate of vIPAG GABAergic neurons. (a) Example unit with low baseline activity (2.0 spikes/s, red line). Laser stimulation at 15 Hz results in an average firing rate of 14.3 spikes/s (blue line). (b) Unit with high baseline firing rate (42.0 spikes/s). Even for this unit, laser stimulation causes an increase in activity to 55.0 spikes/s.

To address more directly whether 20 Hz stimulation does result in an activation (increased firing rate) of vIPAG GABAergic neurons, we analyzed the firing rates of identified vIPAG GABAergic neurons before and during the short laser pulse sequences in optrode recording experiments. For all stimulation protocols (15 Hz, 0.1-0.2s long step pulse, and 30 Hz), the firing rates of all identified units were significantly increased ($p < 0.033$, Wilcoxon signed rank test).

For illustration, Fig. R1 shows the laser triggered firing of two units with low and high baseline activity before, during, and after the 1 s long 15 Hz-stimulation period. The activity of the unit with low baseline activity (2.0 spikes/s, red line) was 14.3 spikes/s during the laser stimulation period (blue line). Importantly, even the unit with a very high baseline activity (42.0 spikes/s) showed an increase in its firing rate during the laser stimulation (55.0 spikes/s). Laser stimulation thus seems to have an “additive” effect, irrespective of the baseline firing rate, suggesting that optogenetic stimulation with 20 Hz indeed results in an activation of vIPAG GABAergic neurons.

2. The number of recorded neurons is low. Optogenetic tagging can indeed result in low yield, but I am surprised at the very low number of non-tagged neurons at the same time, recorded from 6 (!) mice. Is there an explanation? Was this small sample balanced across mice? (How many tagged cells for each mouse?) Nevertheless, I find the combined number of recorded and imaged GABAergic neurons sufficient for the claims.

The numbers of laser-inhibited or unmodulated units for each tested animal are 2, 3, 3, 3, 5, and 6. The locations and number of identified units are depicted for each mouse in Suppl. Fig. 5 (1, 2, 2, 3, 4, and 7 per mouse).

The small number of non-tagged neurons is likely due to a sampling bias in our recording strategy: We were mostly interested in recording GABAergic units, and as the success rate for finding a driven unit is generally very low, we spent the 1-2 hrs on recording only if we found at least one unit that appeared to be activated by the laser pulses. As a result, most of the unidentified units were recorded simultaneously with other laser-driven, putative GABAergic units. We have added this explanation in the Methods section (p. 25, lines 4–8).

3. From Fig.S1 it appears some optic fiber implants were actually below the injected site. Was there an effect on sleep states in these mice as well? I am a bit surprised that unilateral inhibition was sufficient to generate strong effects. Therefore my broader question is how consistent these optogenetic effects were across animals? The authors only show averages.

Figure R2. Effect of activation of vIPAG GABAergic neurons in single mice. (a) Percentage of REM (top), wake (middle), and NREM sleep (bottom) for each single mouse (n = 12). Each row corresponds to a single mouse and color-codes the brain state percentage averaged across single trials before, during, and after laser stimulation (300 s, 20 Hz). (b) All laser stimulation trials from 12 mice. Each row represents the color-coded brain state before, during, and after laser stimulation. Each bracket on the right indicates all trials from a single mouse.

The effect of laser stimulation was highly consistent across mice. For demonstration, we plot in Fig. R2a the percentage of REM, wake, and NREM (relative to laser stimulation) for each single mouse. In each of the 12 tested mice, the probability of REM sleep is reduced, while NREM sleep is increased. Additionally, each single laser stimulation trial of each mouse is depicted in Fig. R2b (each row is one trial). Compared to the preceding 10 min period, during laser stimulation, REM sleep is clearly reduced across trials, while NREM sleep is increased.

A reason why optogenetic activation also works for the mice in which the optic fiber was implanted at a comparably deep position (e.g. Mouse No. 10 corresponds to the mouse with “deepest” optic fiber) could be that tissue scatter of the light emitted by the optic fiber activated some neurons more dorsal to the fiber tip, and/or the light stimulated axons fibers or dendrites of the infected neurons.

Minor points:

1. p.8. "unidentified cell population contained a lower percentage of REM-off neurons" - although one can count the points in the figures, please quantify.

We state now explicitly in the text the number and percentage of unidentified REM-off cells (p. 9, line 1).

2. The authors did a thorough job answering the critics of reviewers. In this regard I welcome the addition of more data with less processing; nevertheless, raster plots of individual example cells would still increase my enthusiasm about the data. I also suggest incorporating some of the non-normalized firing rate data from the reviewer figures into the supplementary material.

The non-normalized firing rates of vIPAG REM-off neurons during brain state transitions and during inter-REM periods are depicted in Supplementary Figs 7 and 10. In Supplementary Fig. 6g,h, we have included an example of a laser-inhibited unit along with firing rates and raster plots.

Reviewers' comments:

Reviewer #3 (Remarks to the Author):

I have a remaining concern I would like to see properly addressed before I can make a final assessment of this manuscript.

1. Since 'firing rate' quantified in short time bins only reflects the bin size (see peaks of 100-150 Hz in Fig R1 that likely correspond to single action potentials), two example plots with this obvious confound does not eliminate my concern. The point is technical, nevertheless important for the interpretation of the results. Unless there is a thorough treatment of this issue, I cannot vouch for the authors' interpretation of optogenetic stimulation. For example, raster plots should illustrate the authors surprising claim that laser stimulation has an 'additive' effect. Summary statistics of spike counts in sufficiently large windows (e.g. 1 s long stimulation periods vs. baseline) are necessary. The analysis should not only entertain the reviewers but also appear in the manuscript.

3. I find this convincing but again, it should be available to the readers, not only to the reviewers.

(Numbers are according to the original points.)

Reviewer #3

I have a remaining concern I would like to see properly addressed before I can make a final assessment of this manuscript.

1. Since 'firing rate' quantified in short time bins only reflects the bin size (see peaks of 100-150 Hz in Fig R1 that likely correspond to single action potentials), two example plots with this obvious confound does not eliminate my concern. The point is technical, nevertheless important for the interpretation of the results. Unless there is a thorough treatment of this issue, I cannot vouch for the authors' interpretation of optogenetic stimulation. For example, raster plots should illustrate the authors surprising claim that laser stimulation has an 'additive' effect. Summary statistics of spike counts in sufficiently large windows (e.g. 1 s long stimulation periods vs. baseline) are necessary. The analysis should not only entertain the reviewers but also appear in the manuscript.

As pointed out by the reviewer, the instantaneous firing rates shown in **Fig. R1** of our previous revision do depend on the bin size. We therefore also used colored lines to indicate the average baseline firing rate (red line) and the firing rate during the 1-s laser stimulation (blue line), which do not depend on the bin size. As we mentioned before, this increase in firing rate was observed for each unit, irrespective of the baseline firing rate.

We fully agree with the reviewer that this important observation deserves a more thorough representation in the manuscript. We therefore added a raster plot for an example unit with high baseline activity to illustrate that the activity of the neurons is still increased (**Supplementary Fig. 5e**). We also provided a summary comparing the baseline activity with the firing rates during both 15 and 30 Hz stimulation for all identified units (**Supplementary Fig. 5f**). Importantly, each single unit showed a significant increase in firing rate during laser stimulation (15 Hz, $p < 0.04$; 30 Hz, $p < 0.02$; Wilcoxon sign-rank test). The firing rate increase was also highly significant across the population of identified units (15 Hz, $p = 0.0001$, $z = -3.3$; 30 Hz, $p = 0.0001$, $z = -3.8$; Wilcoxon sign-rank test).

3. I find this convincing but again, it should be available to the readers, not only to the reviewers.

As suggested by the reviewer, we have included the previous **Fig. R2** (illustrating the effect of laser stimulation on single mice and trials) in **Supplementary Fig. 1 d,e**.